# Seasonal Effects on the Performance of Finishing Pigs’ Carcass and Meat Quality in Indoor Environments

**DOI:** 10.3390/ani14020259

**Published:** 2024-01-14

**Authors:** Fruzsina Albert, Mária Kovács-Weber, Ákos Bodnár, Ferenc Pajor, István Egerszegi

**Affiliations:** Department of Animal Husbandry Technology and Animal Welfare, Institute of Animal Sciences, Hungarian University of Agriculture and Life Sciences, Páter Károly 1, 2100 Gödöllő, Hungary; fruzsi.albert93@gmail.com (F.A.); kovacs-weber.maria@uni-mate.hu (M.K.-W.); bodnar.akos@uni-mate.hu (Á.B.); egerszegi.istvan@uni-mate.hu (I.E.)

**Keywords:** seasonal effect, pork quality, fattening, heat stress

## Abstract

**Simple Summary:**

In the current market, most retailers in EU countries pay pig breeders primarily for the lean meat percentage of the carcass, which does not agree entirely with the quality of pork meat (such as colour). It is well known that pigs are more susceptible than other livestock species to high environmental temperatures. High temperatures in finishing pigs are associated with reduced performance parameters. This study evaluated the seasonal effect (summer vs. autumn) on performance (live weight, average daily gain) and slaughter traits of finishing pigs (warm and cold carcass weights, trunk length, fat thickness), as well as meat quality parameters of the pork (pH at 45 min and 24 h postmortem, colour, drip loss, thawing loss, cooking loss, shear force, and meat composition). Seasonal differences in performance between summer and autumn were more pronounced when measuring the length of the phase of the average daily weight gain during the trial. There was also a significant difference in trunk length and fat thickness parameters (withers, loin, and mean back fat thickness), L* value, total drip loss, and cooking loss between the two groups. In addition, the seasonal differences affected the meat composition parameters.

**Abstract:**

Most retailers in EU countries pay pig breeders for their animals’ lean meat percentage, which does not align fully with measures of pork quality (such as colour). In this study, we investigated the effects of season (summer vs. autumn) on finishing pigs’ performance, carcass characteristics, and meat quality parameters in 24 slaughter pigs. Growing performance traits (live weights, average daily weight gain), slaughter values (warm and cold carcass weights, trunk length, fat thickness) and meat quality parameters (pH at 45 min and 24 h postmortem, colour, drip loss, thawing loss, cooking loss, shear force, and meat composition) were recorded. Seasonal differences were more pronounced for the initial age, the number of days in the growing-finishing phase, and the average daily gain. There was also a significant difference in the trunk length between groups, the fat thickness on withers and loin, and also in mean fat thickness. A significant difference was found in the case of pH, total drip loss, and meat colour (L*). The intramuscular fat and collagen content of meat was significantly higher in summer; in contrast, the protein content of meat samples was considerably lower in summer. In conclusion, seasonal effects on finishers’ performance, lean meat values, and several meat quality parameters highlight the importance of more profound seasonal settings of climate control to fulfil the progressively changing quantitative and qualitative requests of pork sector participants from farm to fork.

## 1. Introduction

The climatic environment is one of the main limiting factors of pig production efficiency. These are essential factors that influence the production achievements of livestock animals and affect the worldwide livestock population [1]. Adverse effects of environmental temperature, relative humidity, and other climate factors cause heat stress, damage health, and decrease welfare. Finally, if these harmful environmental factors become chronic, they may result in death [1,2]. The animals’ reaction to heat stress causes critical biological changes that contain several behavioural and physiological adaptive feedback mechanisms in order to maintain homeostasis [1]. Pigs are more susceptible to high environmental temperatures than other livestock species as they can neither sweat nor pant. Heat stress is responded to by a complex set of physiological, behavioural, and anatomical mechanisms designed to promote heat loss to the environment and minimise heat gain from the environment. Pigs have been raised in closed systems due to their high lean meat content and fast growth since the 19th century. These fast-growing pigs produce more heat from their feed intake than the pigs raised before closed-system farming techniques were employed. Combined with confinement, this aspect makes it challenging to manage the heat balance of pigs in intensive systems. The primary consequence of heat stress is that pigs reduce their feed intake in proportion to the increase in temperature, which, in turn, reduces their performance on meat quality [3]. Earlier studies showed that pigs are more susceptible to high environmental temperatures, primarily when combined with high relative humidity [4,5]. Economic losses associated with heat stress in the livestock sector include slower growth, reduced fertility, increased veterinary costs, deterioration in carcass quality and composition, reduced market weight, and decreased animal welfare [5,6,7].

Unfortunately, the adverse consequences of heat stress on animal health and its negative effects on production will increase, especially if the selection for improved production traits is favoured over thermotolerance and climate adaptation [8]. Poultry, pigs, and ruminants are susceptible to heat stress because of their fast metabolism, increased metabolic heat production, rapid growth, and high production levels. Acute heat stress immediately before slaughter increases the lactic acid concentration in muscles, accelerates muscle glycogenolysis, and causes a rapid decrease in muscle pH in the warm carcass in the early postmortem phase [9]. This will result in the pale, soft, and exudative (PSE) meat defect, characterised by a lower water holding capacity (WHC) and commonly observed in pigs and poultry [10,11]. In animals exposed to chronic heat stress, muscle glycogen reserves are reduced, leading to lower lactic acid production, resulting in dark, firm, and dry (DFD) meat with a high final pH and higher WHC, which can also occur in pigs [12]. Despite many technological advances in indoor climate control in livestock buildings in recent decades, the ambient temperature is not always appropriate for the pig’s thermal comfort zone [13]. Previous results suggest that heat stress causes a negative effect on fattening and some meat quality parameters. Still, the impact of season on meat quality traits (such as colour, drip loss, thawing loss, cooking loss, shear force, and meat composition) is less known. Our experiment aimed to investigate the seasonal effects (summer vs. autumn) on finishing pigs’ gaining performance, slaughter, and meat quality parameters.

## 2. Materials and Methods

### 2.1. Experimental Design

The experimental procedures and animal care conditions employed in this study complied with the European guidelines for the care and use of animals in research [14]. This study was conducted at the Research Institute for Animal Breeding, Nutrition, and Meat Science (Herceghalom, Pest County, Hungary; GPS coordinates: 47.49867 N, 18.75412 E).

Twenty-four (Hungarian Large White, HLW x Hungarian Landrace, HL) × (Pietrain × Duroc) four-line hybrid pigs were introduced into the experiment (twelve during the summer period and twelve during the autumn period) in a 50/50 sex ratio from six litters per season (a gilt and a barrow per litter with a mean litter size of 12.3 pigs per sow) until reaching the final weight of 110 kg per pig. The start of the finishing phase in autumn was on 19 September, and in summer it was on 7 May. The external meteorological data recorded during the investigation period are shown in Table 1. Moreover, the max/min temperature in the barn was recorded every day using a plastic max/min thermometer (Kerbl, Buchbach, Germany).

The starting live weights of the animals were similar (38.13 vs. 41.02 kg). Regarding the results obtained from the fixed slaughter weights, the animals were slaughtered at almost identical live weights. In the current experiment, a 2-phase feeding program was adopted (Table 2). Feed samples were collected from the feed bag and frozen, and stored at −20 °C until further analysis. Samples from concentrates were analysed for crude protein, digestible energy, ash, calcium, and phosphorus according to the procedure of the Hungarian Feed Codex [15]. In the experiment, the diets were formulated according to the NRC requirements [16].

All pigs were housed individually in pens in a traditional finishing barn. An area of 1 × 1.8 m was provided for each pig. All pens were adjusted with a stainless steel self-feeder and a nipple drinker with ad libitum access. The amount of concentrate offered to the pigs was recorded daily on an individual basis and the refusal the following day. All pigs were housed in a mechanically ventilated building with partially slatted concrete floors. The temperature of the building was maintained using controlled heaters and fan ventilation. Lightning was regulated to 10–12 h of artificial light per day. The pigs’ live weights (LW) were determined at the start and end of the investigation. The average daily gain (ADG) was calculated during the experiment.

### 2.2. Slaughter Procedure

The animals were transported to the low-input slaughterhouse (yearly output: 4000 pigs). The distance between the finishing building and the slaughterhouse was only 500 m. Six pigs in a group were transported from the barn to the abattoir by a tractor with a two-axle open trailer and an average stocking density of 0.5 m^2^/pig. The mean speed of the transportation was 10 km/h and the journey lasted approximately 5 min. At the abattoir, all pigs were immediately unloaded using a metal ramp (5-m length, slope < 15°) without waiting. The mean loading time was 5 min. After loading, animals were weighed and moved to a roofed lairage pen (5 m × 4 m), where all pigs rested for three hours in a group (stocking density was 0.7 m^2^/pig); water was available ad libitum during this period. The ambient temperature was regulated with natural ventilation. During transportation and lairage, plastic sorting panel handled all animals gently and calmly. Animals’ live weights were recorded immediately before slaughter. Electrical stunning was used before bleeding out. Slaughtering of the pigs and processing of the carcasses were conducted according to Hungarian national standard practices. The eviscerated and split carcasses were washed, weighed, and lean meat percentage was measured with a Fat-O-Meat’er (FOM) device (Carometec A/S, Herlev, Denmark). The dressing percentage (the ratio between the weight of warm carcass weight and final body weight at the slaughter) was calculated. pH was determined 45 min postmortem using a pH meter (HI-99163 Food care pH Meter, Hanna Instruments, United Kingdom). Every carcass received a traceability number before it was conveyed into chilling to track its identity. The carcasses were transferred to a cold chamber at 4 °C and refrigerated for 24 h. After 24 h cooling, cold left carcass weight, trunk length, fat thickness (measured on three points), and pH at 24 h postmortem were determined. The fat thickness (on withers, loin, back, and the mean average of these three measurements), and trunk length (obtained with measuring tape) were measured with a stainless steel ruler on the midline of the split carcass in millimetres. Carcasses were weighed again after chilling to determine cold carcass weight.

### 2.3. Meat Quality Analysis

After cold carcass weighing, three slices of pork (*M. longissimus dorsii*, between 12 and 13 ribs) were cut (1 cm width) from the left side of the carcass.

On one slice, the CIELAB system was applied to evaluate meat colour lightness (L*), redness (a*), and yellowness (b*) parameters. Measurement was taken using a Minolta R-410 Chroma meter with 50 mm head (2° standard observer, C light source) (Konica Minolta Sensing, Inc., Osaka, Japan). The colour apparatus was calibrated against a white calibration plate before each reading.

The following formula calculated the total colour difference (ΔE*ab) between groups [17]:ΔE*ab = ((ΔL*)2 + (Δa*)2 + (Δb*))1/2 (1)

The total colour difference (ΔE*ab) was evaluated by the visual perceptibility scale of Lukács [17], where ΔE*ab less than 1.5—not perceptible; ΔE*ab from 1.5 to 3—perceptible; ΔE*ab from 3 to 6—well perceptible; ΔE*ab more than 6—great perceptibility.

A Honikel Test [18] was applied to measure drip loss. The meat samples (a sample weight was approximately 100–125 g) were suspended on plastic hooks and then placed into a refrigerator at 4 °C. The initial weight of the sample and the weight after 24 h (0–24 h), 48 h (25–48 h), and 72 h (49–72 h) were measured before the total drip loss was calculated (0–72 h).

The other two samples were stored frozen (−20 °C) until further analysis. One month later, additional meat quality (thawing loss, cooking loss, WB shear force) and composition parameters were determined in a laboratory.

The second samples were thawed at 4 °C for 12 h and at room temperature (22 °C) for 2 h and then the weight (thawing loss) was measured. Cooking loss was determined as described by AMSA [19]. Briefly, the second meat samples were grilled until the core temperature reached 72 °C in a Cucina HD 2430 contact grill (Philips, Hamburg, Germany). Temperature was measured in the centre of the meat slice with a digital thermometer (DET1R, Voltcraft, Hirschau, Germany). After the heat treatment, the samples were cooled to room temperature (23 °C) and weighed again to obtain the cooking loss. Following this, the test specimens were cut from the slices. The shear force measurements were performed using a TA.XT Plus texture analyser (Stable Micro Systems, Godalming, Surrey, UK) equipped with a Warner Bratzler blade (60° angle, 1 mm thick, shearing speed: 250 mm/min). Shear force was calculated based on the force per unit time (kg) diagram using Texture Exponent 32 software.

The third samples were thawed at +4 °C for 24 h before measurements were taken to test the meat composition values. The slices were homogenised with a hand-held blender (HR1600 Pro Mix Daily Collection, 550 W, Philips, Amsterdam, The Netherlands) and placed in a test vessel. After surface homogenisation, the chemical composition including moisture, intramuscular fat (IMF), protein, ash, and collagen content, was determined using near-infrared spectroscopy (NIR) with a Perkin Elmer DA6200 in percentage terms, such as moisture, intramuscular fat, protein, ash, and collagen content.

### 2.4. Statistical Analysis

Statistical analysis was processed using the SPSS 27.0 (IBM Corporation, Armonk, NY, USA) software package. Shapiro–Wilk’s test was used to test the normality distribution. After that, the homogeneity of variance (F-test), *t*-test, Welch’s *t*-test (if the variables had unequal variances), and Mann–Whitney (when a variable did not meet the normality assumption) tests were performed for each parameter.

## 3. Results

### 3.1. Growth Performance of Finishers

Table 3 presents the statistical data of the finishing pigs’ parameters (*n* = 24). Seasonal differences significantly influenced the initial age, the days in the growing-finishing phase, the weight gain during growing-finishing phase as well as the average daily gain.

The animals’ fattening in autumn started at an average age of 113.75 days, while finishers in summer were an average of 93.5 days old (*p* < 0.001). The autumn-growing animals required on average 74.17 days; in contrast, the summer-growing animals required 91 days to reach slaughter weight (Table 3). The subsequent high ADG in autumn (1009.54 g) resulted in these animals being fattened for significantly less time (74.17 days) than in summer (91 days). The animals were slaughtered at an average age of 184.5 days in summer and 187.92 days in autumn.

### 3.2. Carcass Parameters

The carcass parameters are shown in Table 4.

The seasons did not influence the warm carcass weights and cold left half weights in this experiment. The lean meat percentage and the dressing percentage (S: 81.2% vs. A: 79.1%; *p* > 0.05) were also similar in the two groups. In contrast, the most apparent difference between the summer and autumn experiments was in trunk lengths (*p* < 0.001). The mean values were higher in summer (106.17 cm) than in autumn (100.79 cm). The fat thickness on withers (*p* < 0.001), loin (*p* < 0.05), and the mean value of fat thickness measurements (*p* < 0.05) had significantly increased in autumn finishers compared to those in summer (37.67 vs. 29.75 mm, 26.42 vs. 22.25 mm, and 28.42 vs. 23.42 mm, respectively) (Table 4).

### 3.3. Meat Quality Parameters

The examined meat quality parameters are shown in Table 5.

Summer drip loss results were slightly higher than those in autumn, except for the total drip loss, which was significantly higher in summer (*p* < 0.01). Otherwise, the seasonal differences significantly influenced the rate of cooking loss (*p* < 0.001), and the tendency was similar in drip loss. The autumn group showed only 20.65% cooking loss, while the same value in the summer group was significantly (*p* < 0.001) higher (37.23%). Additionally, a significant difference was evident between the two groups for the L* value (*p* < 0.01). However, thawing loss, shear force, and the a* and b* colour parameters did not differ between the groups. Moreover, the total colour difference (ΔE*ab) between the two groups’ meat samples was 5.22.

The meat composition is shown in Table 6.

Significant differences were measured between the meat composition values. Seasonal changes were observed in the case of protein, intramuscular fat, and collagen contents. For protein (*p* < 0.05), intramuscular fat component (*p* < 0.01) and collagen (*p* < 0.01) measurements were significantly higher in summer, while protein content was lower.

## 4. Discussion

Most EU countries’ pig prices are calculated after carcass grading for lean meat percentage. However, a recent study evaluated consumer preferences for pork in some EU and non-EU countries. It was revealed that consumers’ overall top three motives to buy and eat pork were the quality, price, and taste [20]. The productivity of pig farms depends on several factors: breeds, housing technology, the rotation of finisher barns influenced by day feed-in, time of empty, finishers’ ADG, feed conversion ratio, and, finally, abattoirs’ pricing of slaughter animals [21]. There is a high demand for high-quality and equivalent carcasses from meat processors and consumers throughout the year. Seasons affect pigs’ production efficiency, especially in hot temperatures [22,23]. In the present investigation, the seasons also significantly affected the number of days in fattening, and a significantly higher ADG was observed in the cooler period (Table 3). Čobanović et al. [24] reported that seasons greatly affected the live weight and average daily weight gain; finishers slaughtered in autumn had higher performance. In summer, heat stress can occur, which means that pigs have less feed intake, resulting in lower ADG and taking longer to reach the final slaughter weight by more days. All these phenomena are connected directly with the elevated temperature and are caused by altered intestinal barrier function, inflammatory response, and postabsorptive metabolism [22]. Such marked deviations were observed in this trial in the mean daily feed intake and in the mean calculated feed conversion efficiency in summer 2.14 kg/day and 2.79 kg/kg, and in autumn 2.52 kg/day and 2.50 kg/kg, respectively. The damaged barrier of the intestinal epithelium and oxidative stress can be decreased using elevated Se and vitamin E levels in the ratio [25]. Furthermore, in an intensive system, innovative housing solutions and feeding technologies can also reduce heat stress. Čobanović et al. [2] showed that the slaughter season significantly affected daily weight gain, and, in their research, there were also significant differences in average slaughter age, too. Contrary to this study, our experiment did not detect a seasonal effect on the final age of the slaughtered animals; however, it seems that ADG was constantly higher in lower-average-temperature fattening periods, which is consistent with earlier results [24], showing that daily weight gain in winter was the highest and in summer was the lowest. At the same time, every 1 °C decrease in the environmental temperature increased the pigs’ feed consumption by 35 g/day and increased nutrient requirements, therefore, under colder conditions, metabolic demands also increased [26,27]. Reducing the crude protein content of the feed and increasing the fat content is a solution for reducing the summer loss of appetite in pigs [28]. In the study by Čobanović et al. [24], the season also affected fat thickness. The results showed seasonal influences on fat thickness, as animals incorporate more fat in autumn to protect themselves from the colder temperatures. Our results were consistent with previous reports, where the chronic heat stress during the finishing period (32 °C) resulted in back-fat thickness (−0.3 cm) compared with those maintained at the thermoneutral zone (21 °C) [29,30].

Čobanović et al. [24] found the highest value of hot (86.63 kg) and cold (85.33 kg) carcass weight in autumn, which is not supported by our results. Rinaldo and Mourot [31] found that Large White pigs had reduced feed intake (−13%) and daily weight gain (−12%) during heat stress, higher longissimus dorsi pH (5.71 vs. 5.52), and lower drip loss (20.7 vs. 21.4). Pigs adapt to higher temperatures by reducing the fat layer under the skin, thus facilitating heat loss [8]. As feed intake decreases, muscle glycogen storage decreases, which reduces the glycolytic potential at slaughter, thereby increasing muscle pH [28]. Our experiment also confirmed this.

Carcass weight and lean-meat percentage are beneficial for abattoirs; however, pork quality is necessary for consumers and processors. To date, the relationship between quality and the finishing season has been based on less scientific data. Pork quality encompasses typical traits that ensure meat properties for further processing and storage. Essential features are water-holding capacity (WHC), colour, fat content, composition, and oxidative stability, which are influenced by several breeding, feeding, and technological factors [32]. The WHC of pork is an essential factor because its industrial use is remarkable. Furthermore, drip loss is related to the pH of meat as a faster pH decrease will result in more intense protein denaturation, which leads to poorer water retention and, thus, higher drip loss [33]. The total drip loss and cooking loss data showed significant differences between the seasons. These differences were more intense in summer, negatively influencing pork’s water-holding capacity. The cooking loss depends on the thawing procedure, raw meat quality, and type of cooking process [34,35,36]. Guo et al. [36] compared seven thawing methods and the microwave vacuum thawing (MVT) process resulted in half the thawing loss than conventional and other trial methods. Furthermore, MVT increased the pork WHC by more homogenous water distribution in samples and fewer gaps on their microstructure surface. Decreased cooking loss can be expected by preparation of the samples on the grill (especially contact) compared to oven roasting (high temperature) [34,35]. Our cooking loss results in autumn confirm previous findings, which reported that cooking losses in the meat samples on the grill were 20–27% [34,37].

The colour of pork meat is an important aspect of consumer demand. During this decision, consumers visually assess the freshness and quality of pork [38]. Meat from heat-stressed animals showed lower a* (16.4 vs. 17.9) and higher L* (45.7 vs. 44.4) at 45 min after slaughter, and lower b* (11.4 vs. 12.4) and higher L* (52.3 vs. 50.7) at 24 h after slaughter [39]. In our experiment, the seasons did not significantly affect the a* and b* values. Moreover, the total colour difference (ΔE*ab) between the two groups’ meat samples was 5.22. This value was well-perceptible for consumers according to the visual perceptibility scale of Lukács [17]. Altmann et al. reviewed consumers’ preferences for fresh meat and assessed that more than 50% of participants preferred dark-pinkish colour pork [40]. Čobanović et al. [24] found that the slaughter season significantly affected thawing losses and the L* value, too. They also found significant differences between pH values, drip, and cooking losses. A similar pattern was observed in our experiment, except for the pH2 and thawing loss values; in addition, in the study of Čobanović et al. [7], summer conditions increased the prevalence of PSE meat compared to the winter period, which was similarly observed in our study.

In the present study, the meat’s intramuscular fat and collagen contents significantly increased, while the meat protein content markedly decreased in the summer. Heat stress influences lipid metabolism and lipid storage in growing pigs. In this way, long-term heat stress increases lipid metabolism in adipose tissue [41] and enhances lipid storage and total lipid content in adipose tissue [41,42], resulting in greater back muscle fatness. Moreover, chronic heat stress increased proteolysis and reduced protein concentration in muscle in pigs and broilers [8,43,44]. Elevated temperature can also affect nutrient digestibility, e.g., ileal digestible lysine resulting in lower daily weight gain, protein deposition, and feed conversion rate. With the divergence in the ileal digestible lysine/digestible energy ratio, the body’s fat content increases, i.e., meat quality may be altered [45]. In addition, heat stress altered the collagen content in the belly fat in pigs kept at 32.2 °C for a period of 35 days during the fattening period, increasing the amount of collagen in belly fat compared to fattening pigs kept at 23.9 °C [46]. IMF and collagen play a crucial role in forming pork tenderness. These parameters also depend on the breeds, age of slaughtered animals, and gender (entire male, castrated, gilts) [47,48].

## 5. Conclusions

In conclusion, seasons affected finishers’ growth performance, and the furthermore slaughter and meat quality parameters. Finally, the average slaughter weight and lean meat percentage were similar in both investigation periods, thus, the carcass classification-based price category remained fixed. However, meat quality was inferior in summer, especially regarding the WHC of pork. In this case, both processors and consumers are facing losses while working with such raw materials. Different uses of lower-WHC pork slaughtered in summer are advised, such as dried meat production and grilled dishes (high initial heat) instead of cooked ham or pork roasted in an oven at high temperature. There is a requirement for rapid detection of the lower processing capability of pork, immediately before or after slaughtering, in order to give more profound recommendations for processors and consumers.

## Figures and Tables

**Table 1 animals-14-00259-t001:** Meteorological data for the experimental periods.

Items	Summer			Autumn		
	May	June	July	September	October	November
Average temperature (°C)	18.84	20.56	21.55	16.59	11.81	5.95
Min. temperature (°C)	11.34	14.68	14.62	9.96	5.30	1.76
Max. temperature (°C)	26.36	27.26	28.89	24.52	19.48	10.96
Humidity (%)	68.65	72.73	69.90	73.93	74.84	86.17
Temperature in building (°C)	19–21	24–26	24–26	20–21	19–21	19–21

**Table 2 animals-14-00259-t002:** Composition of the experimental diets.

Items	Phase I (30–70 kg)	Phase II (70–110 kg)
Ingredients (%)		
Corn	38	35.5
Barley	32.61	34.4
Extr. soybean meal 46%	18	11.5
Extr. sunflower meal 34%	6.48	11.5
Sunflower oil	2	1.5
Limestone	0.8	0.9
MCP (monocalcium phosphate)	0.5	0.2
Salt	0.35	0.35
L-Lysine-HCL	0.39	0.35
DL-Methionine	0.12	0.1
Threonine	0.1	0.1
Tryptophan 98%	0.05	0
Aroma	0.1	0.1
Zeolite universal	0	1
Premix *	0.5	0.5
Chemical composition (%)		
Digestible energy (MJ/kg)	14.04	13.50
Crude protein	17.2	16
Ash	5	5.5
Calcium	0.62	0.61
Phosphorus	0.5	0.44
Standardised ileal digestible lysine	0.99	0.84
Standardised ileal digestible methionine	0.38	0.35
Standardised ileal digestible threonine	0.63	0.57
Standardised ileal digestible tryptophan	0.21	0.15

* Premix containing (per kg) for Phase I: Fe 165 mg; Mn 115 mg; Cu 18 mg; Zn 113 mg; Se 0.4 mg; Co 0.2 mg; J 1.2 mg; Vitamin A 10,442 UI; Vitamin D3 1995 UI; Vitamin E 21 UI. For Phase II: Fe 157 mg; Mn 115 mg; Cu 17 mg; Zn 113 mg; Se 0.4 mg; Co 0.2 mg; J 1.2 mg; Vitamin A 10,442 UI; Vitamin D3 1995 UI; Vitamin E 21 UI.

**Table 3 animals-14-00259-t003:** Performance parameters of finishers in different seasons (*n* = 12 per group).

Traits	Season	Mean	Median	SD	*p*-Value
Initial age (d)	S	93.50	93.50	1.17	<0.001
	A	113.75	115.00	2.73	
Initial body weight (kg)	S	41.02	39.90	4.62	N.S.
	A	38.13	38.20	2.87	
Days in growing–finishing phase (d)	S	91.00	91.00	7.31	<0.001
	A	74.17	74.00	5.64	
Age at finishing (d)	S	184.50	184.50	7.40	N.S.
	A	187.92	187.50	6.75	
Final body weight (kg)	S	110.47	107.90	7.17	N.S.
	A	112.75	112.50	6.18	
Weight gain during growing–finishing phase (kg)	S	69.45	69.10	7.62	<0.001
	A	74.62	74.70	7.97	
Average daily gain (g/day)	S	768.23	766.40	111.91	<0.001
	A	1009.54	1007.80	118.05	

S—Summer; A—Autumn; N.S.—not significant.

**Table 4 animals-14-00259-t004:** Carcass parameters between the group of seasons (*n* = 12 per group).

Traits	Season	Mean	Median	SD	*p*-Value
Warm carcass weight (kg)	S	89.73	88.50	5.89	N.S.
	A	89.03	88.70	4.55	
Cold left carcass weight (kg)	S	43.65	42.60	2.69	N.S.
	A	43.22	43.70	2.22	
Trunk length (cm)	S	106.17	107.25	2.25	<0.001
	A	100.79	101.00	2.82	
Lean meat (%)	S	61.55	62.25	2.03	N.S.
	A	59.99	59.70	2.59	
Fat thickness on withers (mm)	S	29.75	29.00	3.84	<0.001
	A	37.67	37.00	6.11	
Fat thickness on back (mm)	S	18.25	18.00	4.99	N.S.
	A	21.17	21.00	5.18	
Fat thickness on loin (mm)	S	22.25	20.00	4.69	<0.05
	A	26.42	25.50	5.16	
Mean fat thickness (mm)	S	23.42	22.00	4.10	<0.05
	A	28.42	27,50	4.92	

S—Summer; A—Autumn; N.S.—not significant.

**Table 5 animals-14-00259-t005:** Meat quality parameters between the groups of different seasons (*n* = 12 per group).

Traits	Season	Mean	Median	SD	*p*-Value
pH1 (45 min after slaughtering)	S	6.18	6.20	0.18	<0.001
	A	6.10	6.05	0.25	
pH2 (24 h after slaughtering)	S	5.62	5.60	0.08	N.S.
	A	5.64	5.62	0.22	
Drip loss 24 h (%)	S	7.44	7.38	2.05	N.S.
	A	6.18	6.39	1.49	
Drip loss 48 h (%)	S	6.34	5.97	1.99	N.S.
	A	5.44	3.89	3.61	
Drip loss 72 h (%)	S	5.34	4.86	2.10	N.S.
	A	4.57	4,28	1.79	
Total drip loss (%)	S	19.12	19.47	4.54	<0.01
	A	16.19	15.32	4.68	
Thawing loss (%)	S	3.57	3.25	1.74	N.S.
	A	4.74	5.21	5.25	
Cooking loss (%)	S	37.23	39.91	8.75	<0.001 ^+^
	A	20.65	20.83	4.62	
Shear force (kg/s)	S	2.63	2.39	0.79	N.S.
	A	2.93	3.08	0.76	
L*	S	71.73	70.88	3.81	<0.01
	A	66.78	67.50	4.97	
a*	S	19.74	19.94	1.08	N.S. #
	A	18.42	20.52	5.47	
b*	S	6.38	6.44	0.53	N.S.
	A	7.36	7.13	2.74	

S—Summer; A—Autumn; N.S.—not significant; ^+^ =Welch’s *t* test; # = Mann–Whitney test.

**Table 6 animals-14-00259-t006:** Meat composition parameters between the groups of different seasons (*n* = 12 per group).

Traits	Season	Mean	Median	SD	*p*-Value
Moisture (%)	S	72.20	72.41	1.19	N.S.
	A	72.08	72.11	0.67	
Intramuscular fat (%)	S	4.79	4.94	1.88	<0.01
	A	3.33	3.50	1.40	
Protein (%)	S	21.03	21.08	0.66	<0.05
	A	23.20	21.96	3.75	
Collagen (%)	S	1.28	2.27	0.15	<0.01
	A	1.13	1.18	0.15	
Ash (%)	S	2.05	2.04	0.11	N.S.
	A	2.11	2.11	0.08	

S—Summer; A—Autumn; N.S.—not significant.

## Data Availability

The data presented in this study are available on request from the corresponding author. The data are publicly.

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
