# Peer review of "Seasonal Effects on the Performance of Finishing Pigs’ Carcass and Meat Quality in Indoor Environments"

_animals, 2024, doi:10.3390/ani14020259_

Round 1

Reviewer 1 Report (Previous Reviewer 2)

Comments and Suggestions for Authors

L 29 – not important, please delete – it was explained in the further text

L 101-102 – please indicate average slaughter weight

L 135 – which national standards?

L167 – frozen under which conditions?

L 171 – under which conditions? – temp., humidity, air velocity?

Why did you not present dressing percentages?

L 303 - WHC of pork is an essential factor because its industrial use is remarkable.??? The sentence is trivial and do not present any important information for readers.

306-314 – why did you choose to explain the CL techniques and result related to them? Why is this important or related with your results?

L 355-359 – this is not clear. How can you conclude something related to the processing technology and suggest to the readers something that is not investigated in your work. Only suggestion, and a wrong one!

Comments on the Quality of English Language

The manuscript requires thorough proofreading by a native person whose first language is English.

Author Response

Manuscript: animals-2807477

Response to Reviewer 1.

We would like to thank Reviewer 1 for taking the time to review our manuscript thoroughly. Below, we indicate our responses to the comments of Reviewer 1. Reviewer’s comments are in italics.

Line 29 – not important, please delete – it was explained in the further text

We deleted it.

Line 101-102 – please indicate average slaughter weight.

The mean slaughter weight was added to the text too.

Line 135 – which national standards?

All these processes were done by the Hungarian national standard.

Line 167 – frozen under which conditions?

All the samples for further investigation were frozen immediately after arriving at the lab and stored at -20°C.

Line 171 – under which conditions? – temp., humidity, air velocity?

The second samples were thawed at 4 ˚C for 12h and at room temperature (22 ˚C) for 2h.

Why did you not present dressing percentages?

Sorry, it was missed before, we added these values to the text.

Line 303 - WHC of pork is an essential factor because its industrial use is remarkable.??? The sentence is trivial and do not present any important information for readers.

Thanks for your remark, we erased this sentence from the manuscript.

Line 306-314 – why did you choose to explain the CL techniques and result related to them? Why is this important or related with your results?

We think that comparing our results in this trait was relevant because it is important for producers and consumers. Furthermore, there was quite a high difference in this parameter between seasons.  

Line 355-359 – this is not clear. How can you conclude something related to the processing technology and suggest to the readers something that is not investigated in your work. Only suggestion, and a wrong one!

The final recommendations were made in concordance with our findings and previous research.

Besides the suggested changes, the whole text was checked by a native English speaker.

Again, we would like to thank Reviewer 1 for the comments and suggestions, which improved the quality and clarity of our manuscript.

On behalf of the co-authors:

Dr. Ferenc Pajor

Reviewer 2 Report (New Reviewer)

Comments and Suggestions for Authors

This study investigated the effects of season (summer vs. autumn) on finishing pigs’ performance, carcass characteristics and pork quality. This topic is interesting, however, appropriate improvement was required.

Major comments

The title must be rephrased to make it more understandable.

There are some language mistakes although the manuscript was generally well written.

A lot of heat stress references were mentioned in the introduction, but the preference temperature for finishers should be discussed. Because it seemed the average temperature in summer was not that high in this study.

Whether the environment could be controlled in the barn?

Which factor would affect the results more, slaughter age or weight? The physiological status could not be same at different age. How to identify the main effect is season but not age?

Minor comments

Table 2: The levels of vitamins and minerals in the premix should be provided.

Author Response

Manuscript: animals-2807477

Response to Reviewer 2.

We would like to thank Reviewer 2 for the review of our manuscript. Below we indicate our responses to the points raised by Reviewer 2. Reviewer’s comments are in italics.

Major comments

The title must be rephrased to make it more understandable.

The title was changed from the original one, and we added a more profound title.

There are some language mistakes although the manuscript was generally well written.

Many thanks for your comment; a native English speaker has proofread the manuscript.

A lot of heat stress references were mentioned in the introduction, but the preference temperature for finishers should be discussed. Because it seemed the average temperature in summer was not that high in this study.

Many thanks for your remarks. You are right; it was not a chronic heat stress situation; however, it was out of the range of the upper thermal neutral zone of finishers in the summer group.

Whether the environment could be controlled in the barn?

Many thanks for the question. Yes, the ventilation and heating elements (in autumn) were controlled in the barn.

Which factor would affect the results more, slaughter age or weight? The physiological status could not be same at different age. How to identify the main effect is season but not age?

Finally, both groups of finishers had similar ages and slaughter weights (Table 3). So, the season had a difference in the performance of the animals. 

Minor comments

 Table 2: The levels of vitamins and minerals in the premix should be provided.

Thanks for your recommendation; we included the premix contents in Table 2.

Besides the suggested changes, the whole text was checked by a native English speaker.

Again, we thank Reviewer 2 for the comments and suggestions that improved our manuscript's quality and clarity.

On behalf of the co-authors:

Dr. Ferenc Pajor

Round 2

Reviewer 1 Report (Previous Reviewer 2)

Comments and Suggestions for Authors

Dear Authors, it is clear that you gave an effort in improving your manuscript. 

According to my opinion the manuscript could be further processed.

This manuscript is a resubmission of an earlier submission. The following is a list of the peer review reports and author responses from that submission.

Round 1

Reviewer 1 Report

Comments and Suggestions for Authors

The manuscript entitled "Seasonal effects on fattening, carcass and meat quality traits of pigs" is important in terms of providing valuable information. I make some recommendations for improving the proposed paper. 

1.     The paper suffers from a poor English structure throughout. The manuscript requires thorough proofreading by a native person whose first language is English.

2.     The novelty of the study needs to be highlighted compared to other similar studies.

3.     In material and method: Heat stress is mentioned in the article, but climate data in the barn was not included in the article. It has been stated that the heating and ventilation inside the barn are controlled (in lines 114-116). How did you create heat stress under these conditions?

4.     Discussion is weak. The discussion needs enhancement with real explanations, not only agreements and disagreements. In this section, it is mentioned that there are differences between summer and autumn growth due to heat stress. However, since we do not have in-barn climate data, we cannot be sure of the source of the difference. Authors should improve it by the demonstration of biochemical/physiological causes of obtained results. Instead of just justifying results, results should be interpreted and explained to appropriately elaborate inferences. The discussion seems to be poor, didn't give good explanations of the results obtained. I think that it must be really improved. Where possible please discuss potential mechanisms behind your observations. You should also expand the links with prior publications in the area but try to be careful to not over-reach.

5.     The conclusion part was similar to the discussion. However, this section is where you should present your conclusions from this article. This part needs to be rewritten.

Comments on the Quality of English Language

The paper suffers from a poor English structure throughout. The manuscript requires thorough proofreading by a native person whose first language is English.

Reviewer 2 Report

Comments and Suggestions for Authors

The manuscript aimed to investigate the effects of season on fattening performance, carcass characteristic and meat quality parameters of pigs.  It was found that  seasons affected the fattening performance, slaughter and meat quality parameters. In general, the manuscript has a potential, but several remarks must be accepted and rephrased in the manuscript. 

Since the pigs were housed indoor, I could suggest the authors to include housing system in the title of the manuscript.

I do not understand what do you want to emphasize when stated that the retailers pays lean meat percentage which is not in line with its meat quality? – why is this important for your manuscript?

Please avoid using the same sentences in the summary and abstract.

L 116  -please explain how the lightning was exactly regulated

L 265-266 – please include some examples of housing solutions and feeding technologies

How could you explain non sig. differences in moisture content and sig. differences in cooking loss  and intramuscular fat content? – please include it in the discussion

How could you explain greater drip loss and cooking loss in summer fattener and not sig. differences in shear force? Please include one paragraph whit discussion related to important trat such as shear force.

What could you recommend to farmers – when is better to fatten pigs, and what could you recommend to buyers? Please specify some solutions for them. You mentioned that some seasonal effect could be improved with some housing conditions – please give some specific suggestions and explain them in discussion.

Conclusion  - it is not clear..again you mentioned your results..nothing for exact conclusion…please conclude!

Comments on the Quality of English Language

The English quality in some sentences must be improved (e.g., L 292-293,..). Some sentences are not clear, such as l 289, l 110, l 66, etc. 

Reviewer 3 Report

Comments and Suggestions for Authors

Comments, review of manuscript id: Animals-2576816.

Title: ”Seasonal Effects in Fattening, Carcass and Meat Quality Traits of Pigs

The manuscript describes the effect of season (summer or autumn season) on production traits, carcass traits and meat quality traits of growing/finishing pigs. Although this is an interesting topic to study, the manuscript in the present form is not good enough to be published. The English writing is not of the quality for a scientific publication, and is now more like a listing of the results rather than a manuscript to be considered for publication. A complete rewriting of the manuscript is needed.

The manuscript in the present form should be rejected.

Below I give some general comments to the manuscript, however, in detail comments will not be given, as this will be a complete rewriting of the manuscript. Such a rewriting is not the job of the reviewer but must be performed by the authors themselves.

Some general comments.

Comments, in general. This is a study with growing-finishing pigs As I understand, what you describe as “fattening pigs” is the last period of the growing-finishing period, called finishing period for modern crossbreeds of pigs. The term “fattening pigs” is not used anymore for modern pigs, and you mist change this term. You must also make it clear if the season in your study or for the entire growing-finishing period, or only for the finishing period.

Title: The title must be improved. Find a better word than “fattening”. What you mean here is growth performance. See my “Comments, in general”.

Simple Summary, The English writing must be improved.

Abstract. The English writing must be improved. See my comments to your use of the word “fattening”.

Introduction. The introduction must be completely rewritten, and the English writing must be improved. You also have several spelling errors, like for instance “and effect worldwide livestock population” in line 42. I may list ore of these spelling errors, but it is better if you find then yourself. Also, much of the introduction is rather general, please be more specific in your statements.   

Materials and Methods, in general. Improve the English language throughout the “Materials and Methods” and be specific in your writing.

Materials and Methods, 2.1, lines 92 - 95. You are writing ”…temperature in the bard was recorded every day by a plastic max/min thermometer … placed on the wall (at the height of animals) closest to the centre of the barn”. This is to many words to describe how the temperature was measured, and the sentence also has information that is not of importance. Rewrite in a more specific way.

Materials and Methods, 2.1, lines 99 – 105. Please specify clearer how you define the experimental period, and do not use the term “animals were fattened”

Materials and Methods, 2.1, Table 1. The table must be improved. For instance, what do you mean with “precipitation (mm)”. Is this the average precipitation per day in the given period? Also, is the amount of precipitation relevant information?

Materials and Methods, 2.1, Table 2. The sum of the ingredients in diet “Phase I” is 100%, but for diet “Phase II” the sum of ingredients is only 97%. Why? Also, you include “Aroma” in the list of ingredients – what type of “Aroma” did you use?

Materials and Methods, 2.3, line 169. To describe cooking loss as “according to a recommended methos” is not good enough in a scientific paper. Be specific.

Results, in general. Improve the English writing and be specific in the way you write. Write it as a complete and concise description and not just as a listing of the results. And, once again, fin a better word than “fattening parameters”

Results, lines 207 – 211. Also, this part is unclear for the readers, be precise in your description.

Results, lines 200 – 204. This is unclear for the readers, be precise in your description.

Results, lines 217 – 227, lines 235 – 244, and lines 251 - 254. Be precise in how you describe this in English.

Discussion, in general. Improve the English writing. Be precise in your description and remember that the discussion must be relevant to the hypothesis if your study.

Conclusions, in general. Improve your English writing.

Comments on the Quality of English Language

The English writing is nor of a good scientific quality, and must be improved.